# CREB Is Activated by the SCF/KIT Axis in a Partially ERK-Dependent Manner and Orchestrates Survival and the Induction of Immediate Early Genes in Human Skin Mast Cells

**DOI:** 10.3390/ijms24044135

**Published:** 2023-02-18

**Authors:** Kristin Franke, Gürkan Bal, Zhuoran Li, Torsten Zuberbier, Magda Babina

**Affiliations:** 1Fraunhofer Institute for Translational Medicine and Pharmacology ITMP, Immunology and Allergology IA, 12203 Berlin, Germany; 2Charité—Universitätsmedizin Berlin, Corporate Member of Freie Universität Berlin and Humboldt Universität zu Berlin, Institute of Allergology, Hindenburgdamm 30, 12203 Berlin, Germany

**Keywords:** mast cells, skin, CREB, SCF, KIT, ERK1/2, survival, apoptosis

## Abstract

cAMP response element binding protein (CREB) functions as a prototypical stimulus-inducible transcription factor (TF) that initiates multiple cellular changes in response to activation. Despite pronounced expression in mast cells (MCs), CREB function is surprisingly ill-defined in the lineage. Skin MCs (skMCs) are critical effector cells in acute allergic and pseudo-allergic settings, and they contribute to various chronic dermatoses such as urticaria, atopic dermatitis, allergic contact dermatitis, psoriasis, prurigo, rosacea and others. Using MCs of skin origin, we demonstrate herein that CREB is rapidly phosphorylated on serine-133 upon SCF-mediated KIT dimerization. Phosphorylation initiated by the SCF/KIT axis required intrinsic KIT kinase activity and partially depended on ERK1/2, but not on other kinases such as p38, JNK, PI3K or PKA. CREB was constitutively nuclear, where phosphorylation occurred. Interestingly, ERK did not translocate to the nucleus upon SCF activation of skMCs, but a fraction was present in the nucleus at baseline, and phosphorylation was prompted in the cytoplasm and nucleus in situ. CREB was required for SCF-facilitated survival, as demonstrated with the CREB-selective inhibitor 666-15. Knock-down of CREB by RNA interference duplicated CREB’s anti-apoptotic function. On comparison with other modules (PI3K, p38 and MEK/ERK), CREB was equal or more potent at survival promotion. SCF efficiently induces immediate early genes (IEGs) in skMCs (*FOS*, *JUNB* and *NR4A2*). We now demonstrate that CREB is an essential partaker in this induction. Collectively, the ancient TF CREB is a crucial component of skMCs, where it operates as an effector of the SCF/KIT axis, orchestrating IEG induction and lifespan.

## 1. Introduction

Mast cells (MCs) are principal effector cells of IgE-mediated type-I hypersensitivity, encompassing allergic rhinoconjunctivitis, allergic asthma, food allergy and anaphylaxis [1,2,3,4]. In the skin, MCs can initiate inflammatory circuits and mediate itch sensations through operating units with sensory neurons [5,6,7,8,9,10,11,12].

Skin MCs (skMCs) orchestrate innate and adaptive immune responses against intruding pathogens [13,14]. They are also involved in a plethora of chronic inflammatory skin disorders such as atopic and contact dermatitis, psoriasis, rosacea and urticaria, where MCs are overabundant and/or hyperactive [15,16,17,18,19,20,21,22,23].

The stem cell factor (SCF)/KIT (“kitten”, also known as c-Kit or CD117) axis represents the most significant receptor tyrosine kinase network of the lineage involved in the induction of pro-survival and function-molding signaling pathways [24,25,26,27,28]. When abnormally activated or mutated, KIT are oncogenic and somatic gain-of-function mutations (especially the D816V mutation), which are causal for systemic mastocytosis [17,24,27].

In our recent report on proteome-wide phosphorylation changes downstream of the SCF/KIT axis in skin-derived MCs, we confirmed well-established modules and identified novel signaling components [29]. Among the regulated transcription factors (TFs) were lineage-identifying entities (e.g., MITF (Melanocyte Inducing Transcription Factor, Microphthalmia-Associated Transcription Factor), GATA2 (GATA Binding Protein 2) and STAT5 (Signal Transducer and Activator of Transcription 5) [30,31,32,33,34,35]), but also general ones such as CREB (cAMP response element binding protein). CREB is one of the oldest TFs from an evolutionary standpoint [29]. It is a prototypical stimulus-inducible TF, whose most significant post-translational modification is phosphorylation at Ser-133. This modulation enhances CREB’s transactivation potential through the recruitment of coactivators to the transcriptional machinery [36,37,38].

Despite CREB’s well-defined role in many cell types and species, surprisingly little information is available on MCs. Research on the lineage has been limited to in vitro generated MC models or cell lines. It has focused on inflammatory mediators downstream of G-protein coupled receptors (GPCRs), infectious agents, FcεRI and other stimuli [39,40,41,42]. In these contexts, CREB chiefly acts as an activator, but was also found to interfere with cytokine generation, e.g., downstream of GPCRs activated by ATP/ADP (adenosine triphosphate/adenosine diphosphate) [39]. To the best of our knowledge, CREB’s contribution to fundamental MC programs such as survival support has not been investigated. CREB’s role in those adaptations that are elicited by the SCF/KIT axis in MCs are likewise ignored.

However, CREB orchestrates survival programs in other cells, especially neurons, thereby mediating neuroprotection and assisting in memory formation [43,44,45]. Survival promotion by CREB maintains neural stem/progenitor cells, while hyperactivity can be oncogenic [46]. Likewise, CREB has been linked to hematopoiesis, where its activation contributes to growth-factor-dependent cell survival [36]. Again, overexpression can be associated with leukemic transformation [36]

CREB’s significance in the neuronal and hematopoietic systems is perfectly in accord with its expression pattern; though believed to be ubiquitously expressed, recent body-wide endeavors revealed substantial variation across cell types. Indeed, expression was most pronounced in the hematopoietic and nervous systems, but weaker in mesenchymal and epithelial cells [47,48]. MCs are myeloid cells, and we found with the FANTOM5 (Functional Annotation of the Mammalian Genome) consortium that skMCs are among the highest expressors of CREB body wide. However, CREB function has not been studied in primary dermal MCs. This gap prompted us to investigate the magnitude of CREB’s contribution to KIT-initiated programs. We report that CREB is constitutively nuclear in skMCs, where it is rapidly phosphorylated by SCF in a KIT-kinase and a partially ERK1/2 (Extracellular Signal-Regulated Kinase ½)-dependent manner. CREB activation crucially contributes to skMC survival and is an important intermediary in the induction of several immediate-early genes (IEGs). Thus, CREB overactivity in MCs may be a factor that contributes to MC hyperplasia observed in and partially causal for inflammatory dermatoses.

## 2. Results

### 2.1. SCF-Triggered KIT Activation Increases CREB Phosphorylation

CREB phosphorylated at Ser-133 was found in skMCs at baseline, but phosphorylation greatly increased following SCF stimulation in a time-dependent manner, reaching a maximum after ≈ 8 min which was maintained for 1 h (Figure 1a). SCF-triggered phosphorylation depended entirely on the intrinsic KIT kinase activity, since it was suppressed by imatinib mesylate (Figure 1b). Interestingly, baseline phosphorylation was reproducibly enhanced by the same substance, suggesting that spontaneous pCREB (phosphorylated CREB) occurrence was not only independent of spurious KIT activity, but even negatively affected by KIT (Figure 1b). We conclude that the SCF-elicited change in CREB phosphorylation expectedly requires intact KIT kinase function, whereas unknown processes lead to pCREB formation in the absence of a defined stimulus, a process negatively affected by KIT kinase.

### 2.2. SCF-Induced CREB Phosphorylation Depends on ERK1/2 Activity but Not on Other MAPKs (Mitogen-Activated Protein Kinases), PKA (Protein Kinase A) or PI3K (Phosphatidylinositol-4,5-Bisphosphate 3-Kinase)

CREB can be phosphorylated by various kinases, including PKA, ERK1/2 and p38 (protein kinase of *38 kilodaltons)*. We therefore set out to investigate which kinase(s) activated via the SCF/KIT axis are responsible for CREB activation. As shown in Figure 2a, interference with ERK1/2 substantially reduced CREB phosphorylation, without abolishing the process completely, while PI3K made no contribution. Similarly, inhibition of PKA had no effect (Appendix A).

The MAPKs p38 and JNK (JUN N-Terminal Kinase) were likewise not involved in KIT-induced phosphorylation of CREB (Figure 2b). In line with the ERK1/2 inhibitor, suppression of the upstream kinase MEK1/2 (MAPK/extracellular signal-regulated kinase 1/2) interfered with CREB phosphorylation, supporting the significance of the MEK/ERK module (Appendix A). As expected, suppression of either the STAT3 (Signal Transducer and Activator of Transcription 3) or STAT5 function did not impede the modification of CREB (Appendix A).

Collectively, among important signaling cascades activated by KIT, only the intrinsic KIT kinase and the MEK/ERK module could be identified as contributors to CREB phosphorylation in SCF-activated skMCs.

### 2.3. CREB Is Phosphorylated Downstream of the IL-33/ST2 Axis

IL-33 was recently reported to elicit CREB phosphorylation in murine bone-marrow derived mast cells (BMMCs) [49]. Human skMCs express vast amounts of ST2 (Suppression of Tumorigenicity 2, IL33 receptor) [50] and respond vividly to IL-33, which differentially modulates their function and positively impacts survival [51,52,53]. We therefore asked whether IL-33 leads to CREB phosphorylation in skin-derived MCs. Indeed, CREB was modulated upon IL-33 stimulation to a similar extent as that following SCF treatment (Appendix A). Combining SCF with IL-33 did not increase CREB phosphorylation, suggesting that maximum phosphorylation is achievable by a single growth/survival factor (Appendix A).

### 2.4. CREB Is Constitutively Present in the Nucleus Where it Incurs Phosphorylation following Activation

CREB is a nuclear protein associated with DNA binding motifs even prior to activation. We investigated whether this distribution can be confirmed in skMCs. Indeed, pCREB was prominent in the nucleus and barely detectable in the cytoplasm. The same applied to total CREB, which was below detection in the cytoplasm, and only detectable in the nucleus (Appendix A). As in whole-cell lysates, phosphorylation was found at baseline in the nuclear compartment, yet it was further boosted by SCF (Figure 3, upper panel). A comparable pattern was found when SCF was replaced by IL-33 (Appendix A).

The pattern contrasted with that of STAT5, which was likewise phosphorylated following SCF administration, yet no pSTAT5 was detected in the absence of stimulus in either compartment (Figure 3, center). After stimulation, pSTAT5 was found in the cytoplasm and nucleus. The ratio of nuclear/cytoplasmic pSTAT5 increased with time, being most pronounced after 60 min (Figure 3, center). Nuclear pSTAT5 and total STAT5 (Appendix A) showed matching patterns and kinetics, suggesting that only the phosphorylated form will appear in the nucleus, as expected.

### 2.5. ERK Is Constitutively Present in the skMC Nucleus Where it Becomes Phosphorylated following KIT Activation

The ERK module is most potently activated by SCF in skMCs [29]. In many cells, ERK activation is followed by nuclear translocation [54]. We therefore asked whether this phenomenon also occurs in skMCs and were surprised to find that this was not the case. In fact, ERK was found in both compartments, though larger amounts were detected in the cytoplasm irrespective of stimulation (Figure 4). Following SCF administration, ERK became phosphorylated in both locations (Figure 4). This indicates that in contrast to other cell types [54], ERK does not change location following skMC activation, but becomes phosphorylated in situ.

### 2.6. CREB Orchestrates skMC Survival Downstream of KIT

CREB has been associated with increased survival of hematopoietic progenitor and myeloid cells [36,55], as well as neurons [43,44,45], with which MCs share several aspects [56]. Since the SCF/KIT axis is crucial for MC maintenance, we asked whether CREB is part of the survival promoting machinery downstream of KIT.

Cells were pre-treated with the selective CREB inhibitor (CREBi) 666-15 at different concentrations and stimulated with SCF for 2 d, after which time cell recovery was determined. As shown in Figure 5a, CREBi dose-dependently reduced the number of identifiable cells with a maximum effect at 5 µM. We then compared CREBi with inhibitors targeting crucial kinases (PI3K, p38 and ERK) and found that across the distinct signaling components, CREB’s action was most pronounced (Figure 5b).

Automatic cell counting reliably determines the number of cells still identifiable as such, but it cannot distinguish between viable and apoptotic cells. Therefore, cell death was directly measured by the YoPro/PI method, which consistently detects apoptotic skMCs by flow cytometry [57,58]. For this purpose, cells were not gated, since the entire population needed to be considered to faithfully capture the percentage of viable cells (plots shown in Appendix A). Cells without re-addition of fresh SCF were also used for further comparison (last addition of SCF 3–4 days earlier). Indeed, inhibition of CREB increased the proportion of non-viable cells (Figure 5c,d). On comparison with other inhibitors, CREBi was again most potent. The effects were especially pronounced when SCF was freshly re-added after CREBi administration (Figure 5d, upper panel), suggesting that CREB was a crucial effector of the SCF/KIT axis. Figure 5c summarizes the two conditions performed on two individual cell cultures each. Figure 5d shows representative dot plots pertaining to the two conditions. The FSC (forward scatter)/SSC (side scatter) plots correspond with the YoPro/PI staining (Appendix A).

We used RNA interference to validate the results. For cell recovery experiments, cells were treated with siRNA (small interfering RNA) for 3 d and cells were quantitated by automatic counting. Indeed, CREB-siRNA had a negative impact on the cell numbers recovered (Figure 6a). In a second series, cells were treated with the respective siRNAs (in the presence of SCF to guarantee survival during the knockdown) for 2 d, then split and cultured for an additional 2 d with or without re-addition of fresh SCF. In both settings and the two independent experiments, the knockdown of CREB negatively affected the proportion of viable MCs (Figure 6b or Figure 6c). Figure 6b gives the cumulative results of the two experiments under both conditions, while in Figure 6c, dot plots of the two experiments and both conditions are depicted (left: without re-addition, right: with re-addition).

In summary, interference with CREB by means of siRNA perfectly confirmed the outcome of pharmacological inhibition (Figure 5), identifying CREB as a crucial element of the survival machinery of skMCs.

### 2.7. CREB Is Indispesable for The induction of Immediate Early Genes (IEGs)

We recently reported that several IEGs are effectively induced in skMCs upon SCF stimulation [29]. Here, we asked whether CREB was an intermediary in this induction. As expected, *FOS*, *NR4A2* and, to a lesser extent, *JUNB* were potently induced by SCF activation. Inhibition of CREB efficiently interfered with this induction, especially of *NR4A2* and *FOS*, while effects on *JUNB* (which was induced to a lesser degree) did not reach significance (Figure 7a). Interestingly, a slight increase in *NR4A2* and *FOS* expression was noted after CREBi treatment in the absence of SCF (Appendix A). Note that expression of these genes is minuscule at baseline (compare the y-axis scale in Figure 7a with the one in Appendix A), as it typically requires a potent stimulus, yet stabilizing effects of CREBi were significant all the same. In contrast, anti-apoptotic *MCL1* [57,58], which is not considered an IEG but is likewise activated by SCF, was not influenced by CREB activity (Figure 7a, right). Likewise, *IL8* and *TNF*, cytokine transcripts induced by SCF in skMCs [29], were not affected by CREBi (Appendix A). 

We used RNA interference to validate the results. Indeed, in both independent experiments (reaching a knockdown efficiency of ≈ 50% or higher), a reduction in CREB potently attenuated the SCF-dependent induction of *NR4A2*, *FOS* and, to a slightly lesser degree, *JUNB* (Figure 7b). Again, the knockdown did not inhibit SCF-elicited *MCL1* expression (Figure 7b, right). We conclude that CREB is a crucial component of the transcriptional machinery specifically controlling the induction of stimulation dependent IEGs.

## 3. Discussion

The evolutionarily ancient TF CREB is highly expressed by skMCs together with several other myelocytes such as neutrophils [47,48,50]. Its association partners CREB binding protein (CREBBP) and E1A Binding Protein P300 (EP300) (both with histone acetyltransferase activity) are also enriched in MCs in comparison with other skin constituents [59]. Therefore, while CREB can be considered ubiquitous, the precise expression levels of its network components differ across cell types. Despite preferential expression in skMCs, CREB’s regulation and function in primary cutaneous MCs has not been investigated, perhaps because of the difficulty in obtaining the required cell numbers for analysis. Our current manuscript is meant as a starting point to close this gap.

SCF is the principal growth factor of human MCs organizing and modulating various aspects of the cells’ biology [24,25,26,27,28]. The manifold modifications elicited by SCF are accompanied by massive changes in the phosphorylation of the MC proteome. In fact, roughly 5400 out of 10,500 phosphosites incur changes following SCF stimulation [29]. We identified the ERK module as most potently activated downstream of KIT, including its upstream kinases and downstream substrates [29]. Here, we demonstrate that the crucial phosphorylation at Ser-133 of CREB likewise requires the MEK/ERK module in skMCs. CREB can be phosphorylated by various proteases, including PKA (activated by cAMP), as the name CREB already implies. In our setting, PKA had no role in CREB phosphorylation however (Appendix A). Other signaling components (p38, JNK, PI3K and STATs) had likewise no role in this post-translational modification. While the MEK/ERK module significantly contributed to CREB phosphorylation, it was not responsible for the entire pCREB generated after KIT activation, suggesting that other, yet to be uncovered pathways play additional parts. p38 can initiate CREB phosphorylation in different cell types [54]. Regarding MCs, p38α was required for murine BMMC maturation partially through its action on CREB [60], while it was not the missing entity in our physiologic MC subset, in accordance with the relatively weak activation of p38 downstream of SCF/KIT in skMCs [29].

Of interest was the observation that ERK1/2 does not translocate to the nucleus following SCF stimulation, but that a fraction is nuclear already in the steady state. Similar to cytoplasmic ERK1/2, the ERK1/2 present in the nucleus becomes phosphorylated following SCF stimulation. Since ERK1/2 is phosphorylated by MEK1/2, and MEK can reportedly shuttle between the cytoplasm and nucleus [61], we postulate that translocation of MEK (or even an event higher up in the hierarchy) is essential to pERK1/2 appearance in the skMC nucleus and we aim to clarify this in the future.

CREB binds to the so-called CRE (cAMP response element), an 8-nucleotide motif (TGACGTCA) that is usually located ≈ 100 nucleotides upstream of the TATA box of selective promoters [36]; around 4,000 of such sites exist in the human genome [62]. Despite the overall significance of CREB as an ancient, well characterized TF that regulates a great number of promoters, its role in MCs has remained ill-defined with a few exceptions (see the Introduction). In particular, there is, to our knowledge, little evidence of CREB being activated by the SCF/KIT axis in MCs. There are isolated reports on other cells however. For example, in murine hematopoietic stem and progenitor cells, the KIT-RAS-CREB pathway seems to be a major module regulated by microgravity [63]. In addition, pCREB was observed in melanocytes following combined SCF/endothelin or single SCF stimulation [64,65]. The variability of KIT signaling in dependence of the precise cell subset is, however, underlined by the observation that in contrast to MCs, STAT5 is not activated by SCF in erythroid cells, while PKA (phosphorylating CREB) is only activated in the latter [66]. This is quite distinct from skMCs, as shown in the present study and our previous work (i.e., no activation of PKA, but significant activation of STAT5 (Figure 3 and [29])). In keeping with the different upstream kinases is the cell-dependent function of CREB itself. While potentially binding to ≈4,000 promoter sites, only a small proportion of CREB target genes are invariably induced by activated CREB in various cells. This suggests that additional events have to unite with CREB activation to elicit biologically meaningful outcomes [62]. They may encompass cell-specific activatory marks in regulatory elements and/or the recruitment of cell-selective coactivators. Therefore, the determination of CREB-regulated genes for each individual cell type is an important endeavor that could be exploited therapeutically. In fact, the targeting of cell-specific CREB-regulated genes, rather than CREB itself, has been suggested as a way forward in the context of neuropsychiatric conditions [67].

As mentioned, SCF/KIT is arguably the most potent axis to ensure MC survival. We recently reported that in primary skMCs, ERK and PI3K activities mediate anti-apoptosis in a partially redundant manner. Therefore, for a limited time, ERK and PI3K can substitute each other; after extended periods, blocking of either module will result in cell demise, while interference with the two pathways in sync will eliminate skMCs completely [29]. Here, we confirm that skMCs can cope with suppression of either ERK or PI3K for a few days, since survival was only moderately affected by their inhibitors. Interestingly, inhibition of CREB in the same setting led to intensified cell death and poorer cell recovery. Though ERK is involved in CREB phosphorylation, the greater potency of CREB over ERK inhibition may be due to at least two not mutually exclusive reasons: a) ERK is not the only kinase organizing CREB phosphorylation, and therefore the impact of ERKi on CREB function will only be partial. b) ERK has several hundred direct (and even more indirect) substrates [68]; some may act in a survival-promoting manner, others in a survival-attenuating manner. In fact, ERK can also phosphorylate and thereby activate transcriptional repressors [54]. In MCs, ERK has also been described as a negative regulator of lineage attributes such as secretory granule constituents [69].

skMCs are protected from apoptosis by mechanisms that encompass Mcl-1 and Bcl-xl as the downstream effectors [57,58]. Mcl-1 seems to be regulated by STAT5, while Bcl-xl requires JNK function leading to the activation of an as yet to be uncovered TF(s) [58]. We tested whether *MCL1* induction also depends on CREB function, but found that this was not the case. It will be of interest to determine whether other anti-apoptotic proteins or modulators require either CREB itself or one of its downstream effectors such as FOS or NR4A2. In any case, CREB positively regulates survival also in other hematopoietic cells where it can even be a proto-oncogene contributing to myeloproliferative disease [36,55].

We recently reported that several IEGs (*FOS*, *EGR1*, *NR4A2* and *JUNB*) are induced in skMCs following SCF activation in a completely ERK-dependent fashion [29]. Such early response genes are directly induced without the necessity of new protein synthesis and their coordinated induction is common to many modes of cellular activation by endogenous and environmental stimuli, as was also highlighted by the FANTOM5 consortium [70]. IEGs can be induced by distinct pathways, including RhoA, ERK and p38 [71,72]. They arise from poised RNA polymerase II complexes [73].

The best-understood target of ERK in the context of IEG induction is ELK1 (ETS-Like Gene 1), a TCF (ternary complex factor) family co-factor of SRF (serum response factor). Indeed, ERK was described to induce ELK1 also in MCs following stimulation via FcεRI [74] and the modification was likewise observed in our global phosphoproteome for the SCF/KIT axis [29]. Therefore, we assumed that SCF-triggered IEG induction in skMCs is the result of SRF activation due to ERK-mediated ELK1 phosphorylation without the necessity of other ERK-targeted TFs. Conversely, we found herein that CREB is an indispensable component of the IEG-inducing machinery of skMCs. In fact, CREB binding sites are prevalent in IEGs, and cooperation between CREB and SRF has been reported in other cell types as well [36,72,75]. The concerted action of the two TF complexes seems to be particularly relevant in neurons and the brain [72,76,77]. We recently reported that, despite different ontogeny, MCs and neurons share a number of features not found elsewhere in the hematopoietic system [56].

Some newly synthesized IEG products are additional TFs, which ultimately organize the activation of late secondary response genes that drive downstream functions and modulate cell fate decisions. This also applies to the IEGs investigated herein (*FOS*, *JUNB* and *NR4A2*). FOS has a complex role in the regulation of survival versus death in the hematopoietic system [78]. The protein was described as abundant in skMCs [79], but its functional implications remain to be elucidated in these cells. In murine BMMCs, *Fos* is upregulated following Ca++ influx [80], and activation-associated induction was also described in the rat MC line RBL-2H3 [81,82]. However, *Fos* deficiency was not associated with altered MC numbers in mice in vitro or in vivo [82,83]. In human intestinal MCs, IL-4 treatment increased *FOS* expression in a MEK/ERK-dependent manner and obviously a STAT6-independent manner [84]. Knockdown experiments will be required to illuminate FOS involvement in the maintenance and function of human MCs, and the same applies to NR4A2. Indeed, the role of NR4A2 in the regulation of MC survival has not been studied so far in any MC subset, but the TF can either enhance proliferation and survival or induce apoptosis in other systems in a cell-subset-specific manner [85]. In hematopoietic cells, for instance, NR4A2 is essential for the maintenance of neutrophil health and lifespan downstream of PKA activation [86]. Moreover, NR4A2 positively regulates the survival of dopaminergic and other neurons and has therefore been proposed as a neuroprotective target for the treatment of neurodegenerative diseases [87]. Interestingly, NR4A2 can transcriptionally regulate other TFs, but also kinases, including ERK1/2 [85].

Two other interesting observations were made in the absence of a specific stimulus: First, the low presence of pCREB in skMCs at baseline was not only resistant to, but even further stabilized by the KIT kinase inhibitor imatinib-mesylate. This suggests that basal activation is the result of an alternative route that is negatively targeted by KIT. Second, the low expression of IEGs in baseline cells was positively affected by CREB inhibition. This seems to imply that MCs produce their own (SCF/KIT-independent) growth factors, which maintain a low degree of pCREB and IEG transcription. Interestingly, in this KIT-independent pathway, the two are not positively linked, i.e., pCREB does not elicit the low-level IEG transcription, while the same factor becomes a key player under SCF stimulation, possibly because an increased level of phosphorylation is required for IEG regulation. These results demonstrate that CREB is indispensable for the KIT-dependent regulation of IEGs in skMCs, while being a negative regulator of their expression in the absence of a specific exogenous stimulus.

In summary, this study reveals CREB as a crucial intermediary of the SCF/KIT axis that fundamentally regulates skMC survival, probably via the induction of key IEGs.

## 4. Materials and Methods

### 4.1. Cells and Treatments

MCs were isolated from human foreskin tissue as previously described [51]. Each mast cell preparation/culture originated from several (2–10) donors to achieve sufficient cell numbers, as routinely performed in our lab [52,53,57,88,89]. The skin was obtained from circumcisions, with written, informed consent of the patients or legal guardians and approval by the university ethics committee (protocol code EA1/204/10, 9 March 2018). The experiments were conducted according to the Declaration of Helsinki Principles. Briefly, the skin was cut into strips and treated with dispase (26.5 mL per preparation, activity: 3.8 U/mL; Boehringer-Mannheim, Mannheim, Germany) at 4 °C overnight, the epidermis was removed, the dermis was finely chopped and then digested with 2.29 mg/mL collagenase (activity: 255 U/mg; Worthington, Lakewood, NJ, USA), 0.75 mg/mL hyaluronidase (activity: 1000 U/mg; Sigma, Deisenhofen, Germany) and DNase I at 10 µg/mL (Roche, Basel, Switzerland). Cells were filtered stepwise from the resulting suspension (100 and 40 µm strainers, Fisher Scientific, Berlin, Germany). MC purification was achieved by anti-human c-Kit microbeads (#130-091-332) and an Auto-MACS separation device (both from Miltenyi-Biotec, Bergisch Gladbach, Germany), giving rise to 98–100% pure preparations (FACS double staining of KIT/FcεRI (anti-FcεRI eBiosciene #11-5899-42), Fisher Scientific; anti-CD117 Miltenyi-Biotec # 130-111-593) and acidic toluidine blue (Sigma) staining, 0.1% in 0.5 N HCl (Fisher Scientific), as described previously [90,91].

MCs were cultured in the presence of SCF, and IL-4 was freshly provided twice weekly when cultures were re-adjusted to 5 × 10^5^/mL. MCs were automatically counted by CASY-TTC (Innovatis/Casy Technology, Reutlingen, Germany) [88,92]. 

Experiments were performed 3–4 d after the last addition of growth factors. For inhibition studies, cells were pre-incubated with 666-15 (CREB inhibitor; 5 µM unless otherwise stated; from Merck Chemicals, Darmstadt, Germany) or SCH772984 (ERK1/2 inhibitor; 10 µM), Pictilisib (PI3K inhibitor; 10 µM), Trametinib (MEK1/2 inhibitor; 10 µM), SB203580 (p38 inhibitor; 10 µM), SP600125 (JNK inhibitor; 10 µM), Pimozide (STAT5 inhibitor; 10 µM) and STAT3-IN (STAT3 inhibitor; 10 µM), all from Enzo Life Sciences, Germany, or imatinib-mesylate (Gleevec, KIT inhibitor; 10 µM, from Biozol Diagnostica, Eching, Germany) or KT 5720 (PKA inhibitor; 2 µM, from Bio-Techne, Wiesbaden, Germany) for 15 min, then stimulated (or not) by SCF (100 ng/mL). IL-33 was purchased from PeproTech (Hamburg, Germany) and applied in a concentration of 20 ng/mL, as described previously [52].

### 4.2. Immunoblot Analysis

After pre-treatment with inhibitors for 15 min and/or stimulation with SCF (100 ng/mL) or IL-33 (20 ng/mL), MCs were collected by centrifugation and immediately dissolved in SDS-PAGE (Sodium Dodecyl Sulphate-Polyacrylamide Gel Electrophoresis) sample buffer and boiled for 15 min (whole-cell lysates). Samples of equal cell numbers were subjected to immunoblot analysis. Membrane blocking was performed in 5% (*w*/*v*) low-fat milk powder (Carl Roth, Karlsruhe, Germany) solution for 30 min. The following primary antibodies were purchased from Cell Signaling Technologies (Frankfurt am Main, Germany): anti-p-CREB (S133, #9198), anti-p-ERK1/2 (T202/Y204, #9101), anti-p-STAT5 (Y694, #9359), anti-t-ERK1/2 (#9102), anti-β-Actin (#4967) and anti-Cyclophilin B (#43603). As a detection antibody, a goat anti-rabbit IgG peroxidase-conjugated antibody was used (Merck, #AP132P). For consecutive development of several molecules on the same membrane, the antibodies (primary and secondary) were removed from the membrane after each detection step by incubation in 0.5 N NaOH (Carl Roth, Karlsruhe, Germany) for 15 min. After each stripping step the membrane was blocked in 5% (*w*/*v*) low-fat milk powder for 30 min (as above), followed by incubation with the next primary antibody. Proteins were visualized by a chemiluminescence assay (Weststar Ultra 2.0, Cyanagen, Bologna, Italy) according to the manufacturer’s instructions. Bands were recorded on a chemiluminescence imager (Fusion FX7 Spectra, Vilber Lourmat, Eberhardzell, Germany). Semi-quantification of recorded signals was performed using the ImageJ software (Rasband, W.S., ImageJ, U. S. National Institutes of Health, Bethesda, Maryland, USA, https://imagej.nih.gov/ij/ (accessed on 30 December 2022), 1997–2018). Individual intensity values for the detected proteins were normalized to the intensity of the housekeeping protein cyclophilin B of the same membrane. Cyclophilin B is an ER-specific cyclophilin [93] and is generally used for whole cell/cytosolic lysates. It was chosen mainly due to its size because it is detected at around 20 kDa and therefore does not interfere with any other antibody.

For protein localization experiments, nuclear and cytoplasmic fractions were prepared using the NE-PER buffer system (Fisher Scientific) according to the manufacturer’s instructions. Protein concentration was determined by the BCA (bicinchoninic acid) method (Fisher Scientific). Equal protein amounts of cytoplasmic (~15 µg) and nuclear fractions (~10 µg) were separated through 4–12% Bis-Tris gels (Fisher Scientific), transferred to PVDF membranes (IB401002, Fisher Scientific) and detection of proteins of interest as well as beta-actin were performed as described above. Beta-actin was chosen as a loading control due to its presence in the cytosol as well as the nucleus [94,95,96].

### 4.3. Cell Recaovery

Differently treated MCs (inhibitors, siRNAs and SCF) were counted after 2–3 d, as specified in the legends according to a routine method [29,57,58,90,91,97]. In brief, cells were diluted 1:200 with Casy buffer (OLS Omni Life Sciences, Bremen, Germany) and cell counting was accomplished by the means of an automatic cell counter and analyzer (Casy Model TTC). Acquisition of the particle diameter allowed to distinguish between cells with damaged and intact membranes. The latter were considered as identifiable cells, and their number served to determine MC recovery in % according to the following formula: (final cell count/plated cell count) * 100. This method does not distinguish between apoptotic and viable cells however.

### 4.4. YoPro-1/Propidium Iodide (PI) Staining

Membrane permeability/apoptosis were determined with the YoPro-1/PI method, as described previously [29,57,58,98]. Briefly, differently treated cells (inhibitors, siRNAs and SCF) were stained with the YoPro^TM−1^ dye (Fisher Scientific) and PI (BD Biosciences, Heidelberg, Germany) for 25 min on ice. Stained cells were measured on a MACSQuant^®^ Analyzer10 (Miltenyi Biotec, Bergisch-Gladbach, Germany) or a BD Calibur (Becton Dickinson, Heidelberg, Germany). Cells were not gated on a particular subpopulation (by FSC/SSC profile), but the proportions of viable cells were determined in the entire population. Data were analyzed with the FowJo analysis software (FlowJo LLC, Ashland, OR, USA). YoPro^TM−1^ staining can detect both early and late apoptotic cells (the latter also positive for PI). The clearly double-negative cells were considered viable and the single- and double-positive cells were considered non-viable and used for calculations. While the appearance of the dot plots varies in dependence of the day of analysis and the flow cytometer, the fully viable population can be easily detected and gated.

### 4.5. Reverse Transcription-Quantitative PCR (RT-qPCR)

MCs (at 5 × 10^5^ cells/mL) were treated with inhibitors for 15 min prior to SCF addition (100 ng/mL) for 25 min, after which time cells were harvested for RNA extraction. Briefly, RNA was isolated using the NucleoSpin RNA kit from Machery-Nagel (Düren, Germany) following the manufacturer’s instructions. cDNA synthesis (reverse transcription kit from Fisher Scientific) and RT-qPCR were performed using optimized conditions as described elsewhere [51], using materials from Roche (Roche Diagnostics, Mannheim, Germany). The primer pairs are summarized in Table 1. They were synthesized by TibMolBiol, Berlin, Germany. The 2^−ΔΔCT^ method was used to quantify the relative expression levels of the target genes to three reference genes (appearing at the end of Table 1).

### 4.6. Accell^®^ Mediated RNA Interference

A well-established and efficient siRNA method for skMCs was utilized [29,53,57,58,99,100,101,102]. In brief, skMCs were transfected twice (on day 0 and day 1) by CREB-targeting siRNA or control siRNA (each at 1 µM) for a total of 2 or 3 d in Accell^®^ medium (Dharmacon, Lafayette, CO, USA) (supplemented with Non-Essential Amino Acids and L-Glutamine (both from Carl Roth)). For apoptosis measurements, the transfection (2 d) was in the presence of SCF to guarantee survival during transfection before the cells were replated and cultured for a further 2 d in the presence or absence of re-added SCF. For RT-qPCR, cells were stimulated with SCF (100 ng/mL) at harvest or PBS as control for 25 min.

A CREB1-specific siRNA (“smart pool of 4”) was used (E-003619-00-0050, Dharmacon); due to profound cytotoxicity in skMCs of the commercial non-targeting control obtainable from Dharmacon, a pool of several siRNAs, which do not interfere with MC survival and/or whose targets are barely expressed by MCs, was used for control purposes. A further control, run in parallel, did not contain siRNA but was otherwise treated identically.

### 4.7. Statistics

Statistical analyses were carried out using PRISM 8.0 (GraphPad Software, La Jolla, CA, USA). Comparisons between two groups were performed using the paired Student’s *t*-test, with a *p* value of less than 0.05 considered statistically significant. For comparisons across more than two groups, an RM one-way ANOVA with a Dunnett’s multiple comparisons test was used, unless specified otherwise in the figure legends. In cases in which the data were not normally distributed, non-parametric tests (Friedmann or Kruskal–Wallis) were applied. 

## 5. Conclusions

Our study identifies the ancient TF CREB as a crucial intermediary in SCF-triggered KIT activation of human skMCs. Thereby, CREB forms part of the MEK/ERK module, the most potently activated of all modules in cutaneous MCs [29]. Interestingly, CREB’s function is critical to ensure the survival of skMCs, whereby its relevance exceeds that of survival-promoting kinases such as ERK and PI3K. CREB is also indispensable for the induction of IEGs downstream of KIT. Since the latter organize manifold processes, it may be expected that CREB is a partaker in various biological programs of skMCs. In summary, CREB’s significance in MCs, especially in human dermal MCs, may be far more prominent than hitherto expected, and it will thus be of interest to further delineate its contributions in activatory routes, such as SCF/KIT and IL-33/ST2, and its cooperation with MC-specific TFs to drive lineage-identifying transcriptional programs.

## Figures and Tables

**Figure 1 ijms-24-04135-f001:**
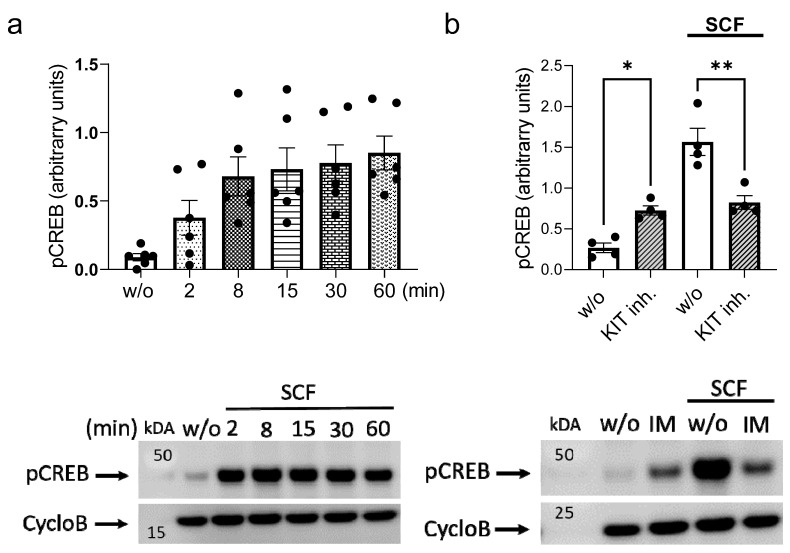
CREB is phosphorylated upon SCF stimulation in a KIT kinase-dependent manner. (**a**) Skin-derived MCs were stimulated with SCF for the indicated times and Ser-133 phosphorylation of CREB was detected by immunoblot in whole-cell lysates. Upper panel: Image J based semi-quantification of the detected signal for pCREB normalized to the housekeeping protein Cyclophilin B (CycloB). Lower panel: Representative time-course. (**b**) Cells were pretreated with imatinib mesylate (KIT inh.) prior to stimulation with SCF for 15 min. Upper and lower panel as in (**a**). Each dot corresponds to one experiment (individual MC culture). * *p* < 0.05, ** *p* < 0.01.

**Figure 2 ijms-24-04135-f002:**
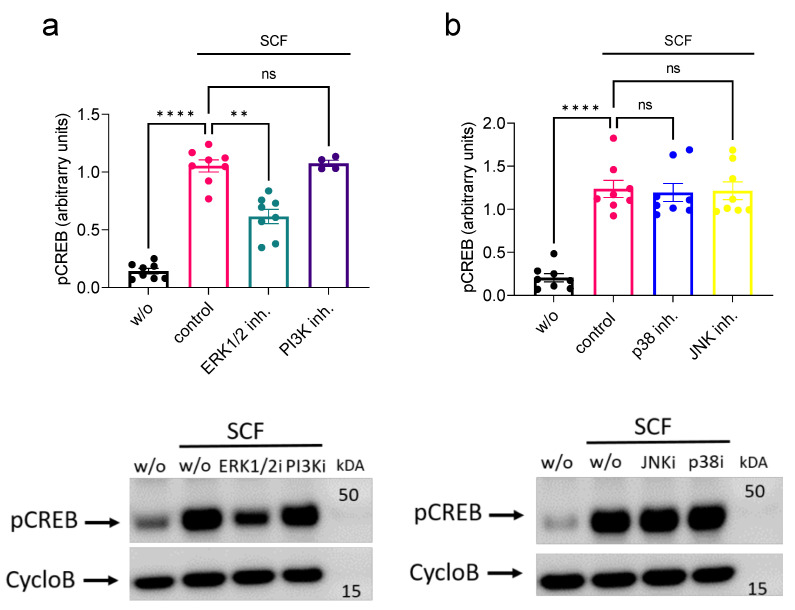
SCF-elicited CREB phosphorylation occurs in an ERK1/2-dependent manner. (**a**) SkMCs were pretreated with inhibitors of ERK1/2 and PI3K and stimulated with SCF for 15 min; phosphorylation of CREB was detected by immunoblot in whole-cell lysates. Upper panel: Image J based semi-quantification of the detected signal for pCREB normalized to the housekeeping protein Cyclophilin B (CycloB). Lower panel: Representative time-course. (**b**) As in (**a**), but inhibitors of p38 and JNK were used instead. Each dot corresponds to one experiment (individual MC culture). ** *p* < 0.01, **** *p* < 0.0001.

**Figure 3 ijms-24-04135-f003:**
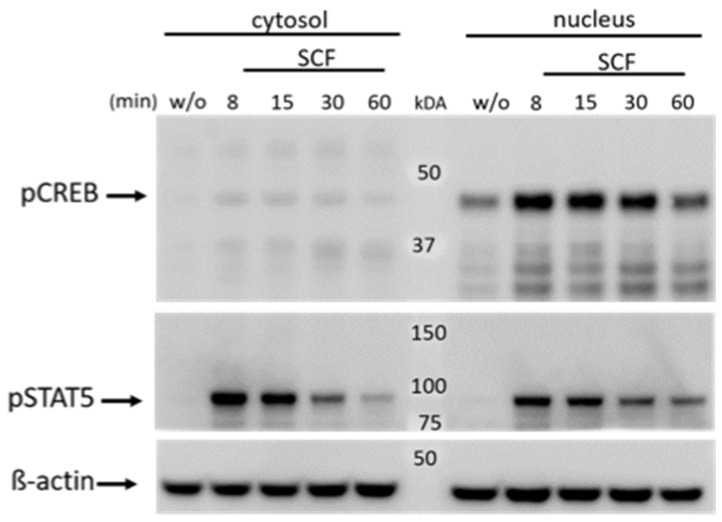
CREB is constitutively located in the nucleus where phosphorylation occurs. MCs were treated with SCF for the times given, cytosolic and nuclear fractions were prepared separately and phosphorylation events were detected by immunoblot. Upper panel: CREB phosphorylated on Ser-133. Center: STAT5 phosphorylated on Tyr-694 given for comparison. Lower panel: β-actin (loading control). One of three experiments with comparable outcomes is shown. For total versus phosphorylated STAT5 and CREB, please refer to Appendix A.

**Figure 4 ijms-24-04135-f004:**
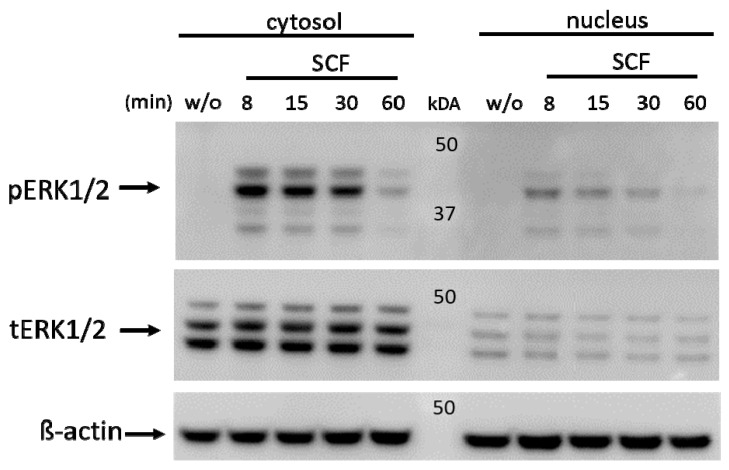
ERK1/2 is constitutively present in the cytoplasm and nucleus; activation-dependent phosphorylation is not associated with translocation. MCs were treated with SCF for the times given, cytosolic and nuclear fractions were prepared separately, and phospho-ERK1/2 (pERK1/2), as well as total ERK1/2 (tERK1/2), were detected by immunoblot. β-actin served as the loading control. One of three experiments with identical outcomes is shown. The same membrane as in Figure 3 is shown.

**Figure 5 ijms-24-04135-f005:**
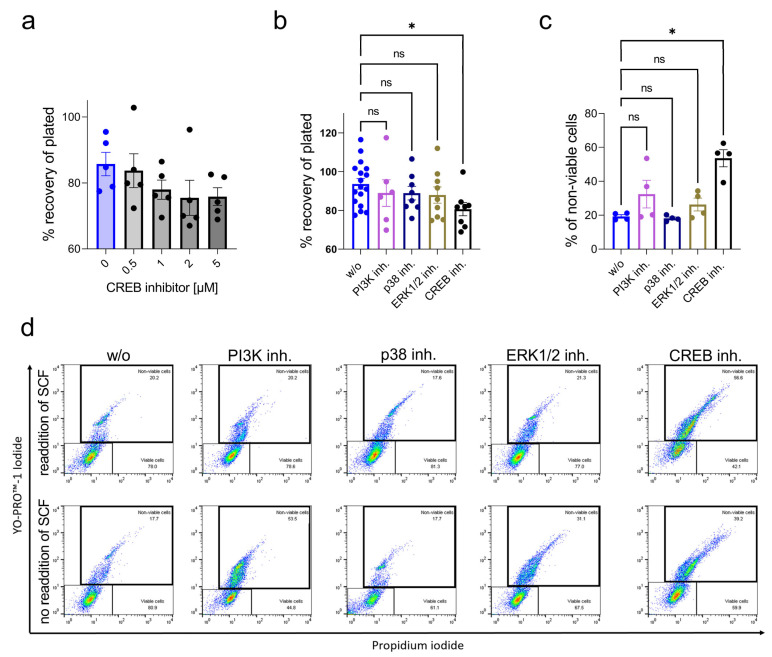
CREB function is required for skMC survival—comparison with other signaling modules. (**a**) SkMCs were treated with different concentrations of the CREB inhibitor for 2 d. Cell numbers were determined and are expressed as % of the original numbers plated. (**b**) As in (**a**), but inhibitors of critical kinases (PI3K, p38 and ERK1/2) were used for comparison. (**c**,**d**) SkMCs were treated with the specified inhibitors for 2 d and viable versus non-viable/apoptotic cells were determined with the YoPro/PI technique. To one portion, fresh SCF was provided directly upon inhibitor pretreatment, the other part had SCF from the previous feeding only (3–4 d earlier). (**c**) Cumulative results. (**d**) Representative dot plots: upper panel with re-addition of fresh SCF, lower panel without. Each dot in (**a**–**c**) corresponds to an individual experiment. * *p* < 0.05 by the Kruskal–Wallis test (**b**) or by the Friedman test (**c**).

**Figure 6 ijms-24-04135-f006:**
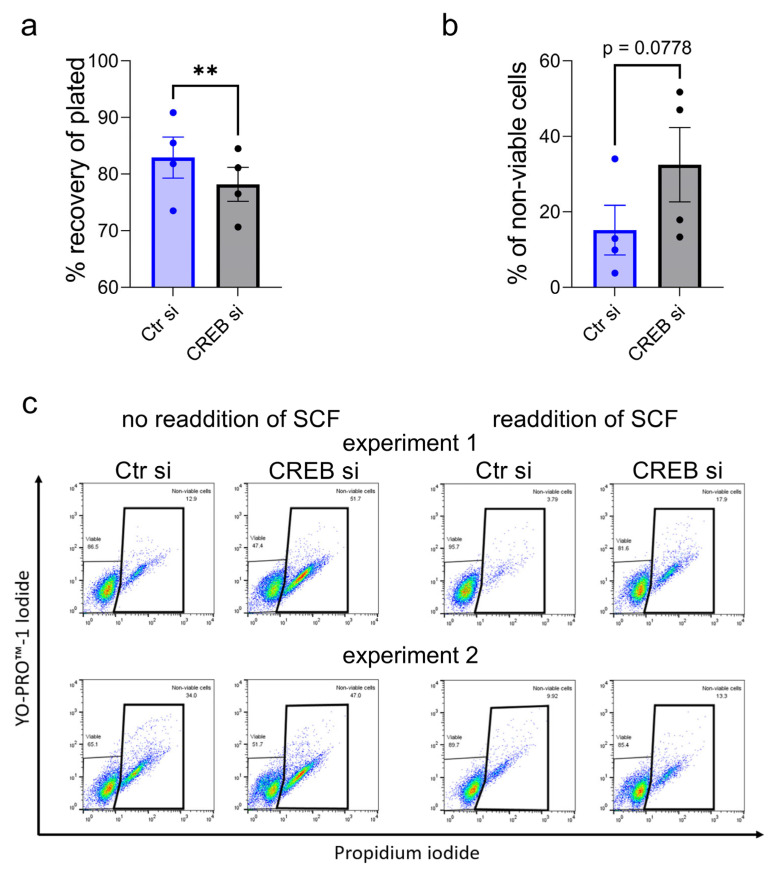
CREB is critically implicated in skMC survival—validation by RNA interference. (**a**) SkMCs were treated with control and selective siRNAs for 3 d. Cell numbers were determined and are expressed as % of the original numbers plated. (**b**,**c**) SkMCs were treated with control and selective siRNAs for 2 d (in the presence of SCF), then replated and kept for 2 more days with or without re-addition of fresh SCF. Viable versus non-viable/apoptotic cells were determined with the YoPro/PI technique. (**b**) Cumulative results of the two experiments run under two conditions each. (**c**) Dot plots of the two independent experiments. ** *p* < 0.01.

**Figure 7 ijms-24-04135-f007:**
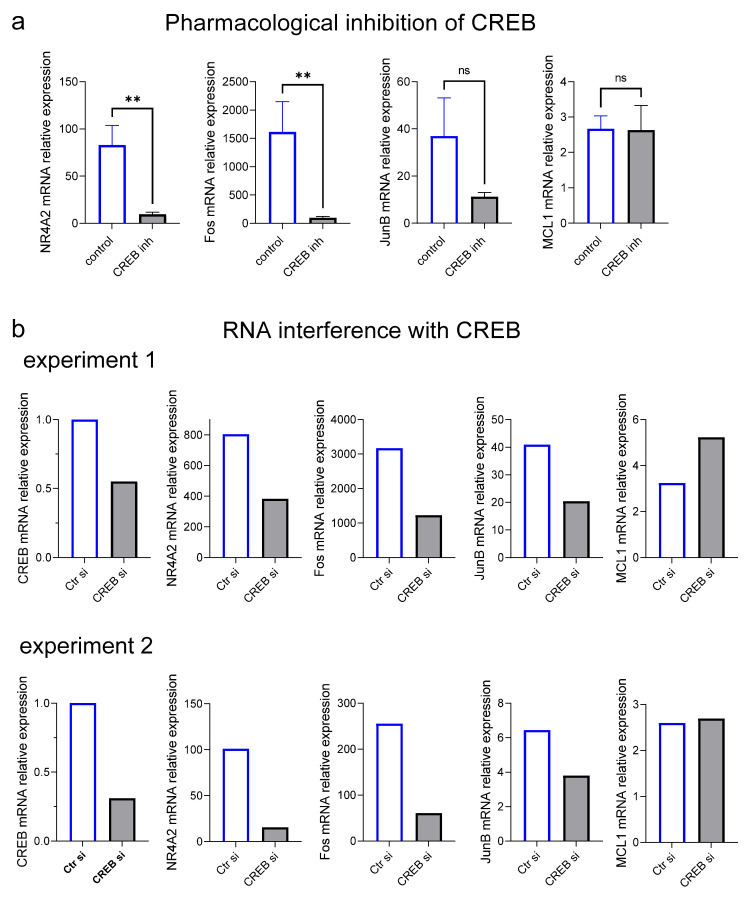
CREB is a critical intermediary in SCF-triggered induction of immediate-early genes but not *MCL1.* (**a**) Skin-derived MCs were pretreated with(out) the CREB inhibitor 666-15, then stimulated (or not) with SCF for 25 min. RT-qPCR was used to quantitate gene expression (normalized to several housekeeping genes). The fold-induction by SCF over control is depicted. Means ± SEM of eight experiments (separate cultures) for IEGs and four experiments for *MCL1* are given. ** *p* < 0.01. (**b**) Skin-derived MCs were treated with CREB-selective versus control siRNA for 48 h, then stimulated as in (**a**) for immediate-early gene quantification. The knockdown efficiency is given on the left-hand side, which was determined in unstimulated cells.

**Table 1 ijms-24-04135-t001:** Primer pairs used for RT-PCR.

Gene	Forward 5′-3′	Reverse 5′-3′
FOS	AGTGACCGTGGGAATGAAGT	GCTTCAACGCAGACTACGAG
NR4A2	TTCTGTAACCCTCCTAGCCC	AGCATGGCCAAACATTTCCC
JUNB	GCCCGGATGTGCACTAAAAT	GACCAGAAAAGTAGCTGCCG
CREB1	GAGAAGCGGAGTGTTGGTGA	TCCGTCACTGCTTTCGTTCA
HPRT	GCCTCCCATCTCCTTCATCA	CCTGGCGTCGTGATTAGTGA
PPIB *	AAGATGTCCCTGTGCCCTAC	ATGGCAAGCATGTGGTGTTT
GAPDH	ATCTCGCTCCTGGAAGATGG	AGGTCGGAGTCAACGGATTT
MCL1	TGCTGGAGTAGGAGCTGGTT	CCTCTTGCCACTTGCTTTTC
TNF	TCTCGAACCCCGAGTGACAA	TCAGCCACTGGAGCTGCC
IL8	ATGACTTCCAAGCTGGCCGTGGCT	CTCAGCCCTCTTCAAAAACTTCTC

* The PPIB gene encodes Cyclophilin B.

## Data Availability

No datasets were generated or analyzed during this study.

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
