# Peer review of "CREB Is Activated by the SCF/KIT Axis in a Partially ERK-Dependent Manner and Orchestrates Survival and the Induction of Immediate Early Genes in Human Skin Mast Cells"

_ijms, 2023, doi:10.3390/ijms24044135_

Round 1

Reviewer 1 Report

Hi,

Even though this article is written by a group with experience in the field, there are a few mentions/omissions.
1) some sentences are a bit too long and would require splitting (hard to read)
2) the text contains a series of abbreviations not explained in the text (it is assumed that the one who evaluates is from the field, and it is no longer necessary?)
3) Some minor grammar mistakes/omissions: passive voice vs. active voice; on/in; employment?? maybe another synonym
4) Some lack of references regarding reagents, equipment, software

Reviewer 2 Report

The authors claim that the ERK-CREB pathway mediates mast cell survival. While this manuscript is well written and of interest for readers in the Mast cell field. However, there are several points which need further explanations and experiments to underpin the authors claim.

Major points:

In Figure 2 the authors show that the ERK inhibitor reduced the SCF induced phosphorylation of CREB. Furthermore, the authors claim that the CREB inhibitor mediates apoptosis. However, the ERK inhibitor did not induced apoptosis in SCF stimulated cells. When ERK mediates CREB activation how do the authors explain that ERK inhibition did not induced apoptosis?

In Figure 3 the authors show and write that CREB is a predominantly nuclear localized protein. However, in Fig. 1 and 2 the authors show very nice pCREB bands and as control blots CycloB, which is a cytoplasmic protein. To this reviewer it is unclear what is shown here. Cytosolic fractions or nuclear fractions or is it just a total lysate containing both fractions. I think that these are total lysates containing both fractions. But in this case the authors must state in the method sections how they did prepare these lysates to lyse the nucleus and they should state this in the legend.

The actin blots in figures 3a, 3b, 4 are identical? Why? When these blots were made from the same lysates it must be written in the legends.

In 3A the total CREB blot must be shown to see whether the sentence “The same applied to total CREB, which was below detection in the cytoplasm, and only detectable in the nucleus (not shown)” is correct. If CREB is a nuclear protein then there should only be very low amounts of it in the cytoplasm.

The authors wrote: “Nuclear pSTAT5 and total STAT5 (not depicted) showed matching patterns and kinetics, suggesting that only the phosphorylated form will appear in the nucleus, as expected”. This is interesting to see and the tSTAT5 blot must be shown in Fig. 3B at least as a control blot.

In den nucleus fractions (Fig 3A, B, Fig 4) the authors only show actin blots as controls for the cytosolic and the nuclear fractions. This is not the appropriated control for nuclear fractions after separation of the cytosolic and the nuclear fraction. A lamin blot must be shown. In contrast to the nuclear fractions there should be no lamin detectable in the cytosolic fraction. Furthermore, a western blot of a typical cytosolic protein must be shown (such a tubulin). The lamin and the tubulin blots will show whether there are cytosolic contaminations in the nuclear fractions or whether there are nuclear contaminations in the cytosolic fractions. Without these controls the blots which are shown in these figures are not convincing.

PI3Ks are known to mediate mast cell survival. In Fig. 5 PI3K inhibition did not induce apoptosis. This is very surprising…..how do the authors explain this?

In Fig 4 the authors show that ERK is mainly located and phosphorylated in the cytoplasm and translocates into the nucleus after activation. However, in the cytoplasmatic fraction there is no decrease of the total ERK blot and there is also no increase of the total ERK blot in the nucleus. This demonstrates no nuclear translocation. How do the authors explain this? Given that the ERK-CREB pathway this is the central claim of the authors for the SCF induced mast cell survival the following question must experimentially shown: Where and how does ERK phosphorylate CREB? These western blots do not underpin these claims. This reviewer assumes that the blots of the nuclear fraction results from contaminations with the cytosolic fraction. The missing tubulin and lamin blots could show whether the preparation of the nuclear fraction is free from such contaminations.

In the Figure 5D the apoptotic cells which are positive for YOPro1 are localized on top of living cells. In contrast to this in Fig. 6C the apoptic cells are localized on the right side of the living cells. Why is the localization of apoptic cells in these two experiments different? How do the authors explain this? This must be clarified. In my opinion the inhibitors and the siRNA induced different ways of cell death.

Furthermore, for the facs analysis the authors must show the gating strategy as a supplement figure.

In Figure 7 the authors show that CREB si RNA reduced the expression of IEGs. However, the CREB inhibitor and the CREB si RNA induced apoptosis. Where do the authors know that the reduced expression of the IEGs did not result from the increased apoptosis induction in treated cells? Less viable cells should also have less IEG expression.

Reviewer 3 Report

Authors addressed the role of the transcription factor CREB as the link in CSF-induced mast cell survival. While this study adds novelty in CREB signaling pathway acting on a central cell type in allergic reactions and convincing evidence is shown, a few minor points might be addressed:

line 95 in my opinion, tittle is confusing, the sentence- inhibition by imatinib mesylate should be removed

line 140 to effect... should be replaced by to affect

lines 225 and 244 authors used the acronym RNAi, however, to make nomenclature consistent siRNA should be written as previously in the manuscript, unless there is a reason why they are using RNAi

In figures 1 and 2 authors conducted experiments using cyclophilin B as housekeeping loading control, unlike other figures where actin was used, is there a particular reason?

line 292 the term famous may be replaced by most common, relevant or frequently used.

Authors addressed the role of ERK on cell survival but not on easrly genes induction, why is that?

lines 350 and 355 authors spell Fos Nr4a2 and sometimes they spell FOS NR4A2, are these genes from different species or is it misspelled?
